# Preparation of Enzyme-Soluble Swim Bladder Collagen from Sea Eel (*Muraenesox cinereus*) and Evaluation Its Wound Healing Capacity

**DOI:** 10.3390/md21100525

**Published:** 2023-10-03

**Authors:** Hangting Li, Jing Tian, Hongjie Cao, Yunping Tang, Fangfang Huang, Zuisu Yang

**Affiliations:** Zhejiang Provincial Engineering Technology Research Center of Marine Biomedical Products, School of Food and Pharmacy, Zhejiang Ocean University, Zhoushan 316022, China; lihangting999@163.com (H.L.); 15533588355@163.com (J.T.); chj19971228@163.com (H.C.); tangyunping1985@zjou.edu.cn (Y.T.)

**Keywords:** *Muraenesox cinereus*, swim bladder, collagen structure, wound healing, immunohistochemistry

## Abstract

In the present research, the enzyme-facilitated collagen from sea eel (*Muraenesox cinereus*) swim bladder was isolated, and the collagen characteristics were analyzed. Then, the collagen sponge was prepared and its potential mechanism in promoting skin wound healing in mice was further investigated. Collagen was obtained from the swim bladder of sea eels employing the pepsin extraction technique. Single-factor experiments served as the basis for the response surface method (RSM) to optimize pepsin concentration, solid-liquid ratio, and hydrolysis period. With a pepsin concentration of 2067 U/g, a solid-liquid ratio of 1:83 g/mL, and a hydrolysis period of 10 h, collagen extraction achieved a yield of 93.76%. The physicochemical analysis revealed that the extracted collagen belonged to type I collagen, and the collagen sponge displayed a fibrous structure under electron microscopy. Furthermore, in comparison to the control group, mice treated with collagen sponge dressing exhibited elevated activities of superoxide dismutase (SOD), catalase (CAT), total antioxidant capacity (T-AOC), and glutathione peroxidase (GSH-Px), and decreased levels of malondialdehyde (MDA), interleukin (IL)-1β, interleukin (IL)-6, and tumor necrosis factor (TNF)-α. The collagen sponge dressing effectively alleviated inflammation in the wound area, facilitating efficient repair and rapid healing of the skin tissue. During the initial phase of wound healing, the group treated with collagen sponge dressing exhibited an enhancement in the expressions of cluster of differentiation (CD)31, epidermal growth factor (EGF), transforming growth factor (TGF)-β1, and type I collagen, leading to an accelerated rate of wound healing. In addition, this collagen sponge dressing could also downregulate the expressions of CD31, EGF, and type I collagen to prevent scar formation in the later stage. Moreover, this collagen treatment minimized oxidative damage and inflammation during skin wound healing and facilitated blood vessel formation in the wound. Consequently, it exhibits significant potential as an ideal material for the development of a skin wound dressing.

## 1. Introduction

Collagen, a significant protein of the extracellular matrix (ECM), is extensively found in various connective tissues such as skin, bone, tendon, and ligament, accounting for around 25–35% of the overall body protein composition [1]. Collagen was widely applied in the fields of food, cosmetics, biomedical, and pharmaceutical industries [2]. In comparison to collagen derived from terrestrial animals, aquatic animal-derived collagen offers unparalleled advantages such as low immunogenicity, reduced risk of disease transmission, and fewer religious and ethical concerns [3]. However, in the course of fishery resource handling, a significant portion of fish by-products is often either discarded or utilized for the production of low-value fishmeal and oil [4]. Therefore, utilizing fish by-products like fish skin, swim bladder, and fish bone as an alternative source of collagen can convert these low-value raw materials into value-added products [5,6], while simultaneously reducing environmental pollution. Furthermore, due to its biocompatibility, non-cytotoxicity, as well as pro-proliferative and antioxidant effects, collagen has become one of the most effective biomaterials for wound healing [7].

Wounds impose a heavy burden on patients and the healthcare system, resulting in significant costs and impacting the patient’s quality of life due to pain, anxiety, and decreased well-being [8]. The process of wound healing is intricate and multifaceted, encompassing inflammation, proliferation, and remodeling, which collaborate harmoniously to facilitate tissue restoration [9]. Multiple studies have shown that collagen, collagen peptides, or collagen composite materials have become the focus of many research studies on wound healing [10,11]. For instance, Chen et al. prepared the collagen sponge from the swim bladder of *Nibea japonica* and demonstrated that the collagen sponge could promote wound healing in mice by reducing serum levels of interleukin (IL)-6, IL-1β, and tumor necrosis factor (TNF)-α at the wound site [12]. Similarly, Sonali Jana et al. combined collagen derived from perch (*Labeo rohita*) skin with bioactive glass containing 3% and 5% Cu^2+^ and Co^2+^, respectively, creating a composite material (Fcol/CuCoBAG) with potential as a low-cost dressing/dermal graft to promote skin wound healing [13]. Additionally, Cheng et al. covalently coupled carboxymethyl chitosan (CMC) and collagen peptide (COP) to prepare a freeze-dried sponge with grafted collagen peptide, which enhanced cell migration and promoted skin regeneration in burn injuries [14]. Collagen, with its remarkable wound healing properties, has the potential to revolutionize wound care and improve patient outcomes, alleviating the burden imposed by wounds on both individuals and healthcare systems.

China boasts abundant fishery resources, among which the sea eel (*Muraenesox cinereus*) stands out as a delicious fish, rich in various nutritional components [15]. Its outer membrane predominantly consists of collagen fiber, which can be extracted to produce collagen and collagen peptides for high-value products [16]. However, there is a scarcity of literature regarding the enzymatic extraction and comprehensive analysis of collagen in sea eel. In this study, pepsin extraction combined with the response surface methodology (RSM) was employed to determine optimal experimental conditions. Besides performing a thorough physical and chemical analysis of the extracted collagen, its application was investigated in a mouse skin wound model to evaluate its impact on wound healing and elucidate underlying mechanisms. These findings aim to establish a robust theoretical foundation for potential applications of collagen in the field of skin wound repair.

## 2. Results

### 2.1. Single Factor Results

The influence of enzyme concentration, hydrolysis time and solid-liquid ratio on collagen extraction rate is illustrated in Figure 1. As depicted, the extraction rate pepsin-solubilized collagen (PSC) sponge initially exhibits an upward trend followed by a subsequent decline. The optimal conditions for achieving the highest extraction rate were found to be a pepsin concentration of 2000 U/g, a hydrolysis period of 10 h, and a solid-liquid ratio of 1:80.

### 2.2. Optimization of Extraction Parameters of PSC Sponge Using RSM

Building upon findings from the single-factor experiment, the RSM was utilized with a Box-Behnken design (BBD) to establish the optimum levels of the three specified factors that notably impact PSC yield. The experimental arrangement and resultant findings have been succinctly compiled in Table 1. Through regression analysis of the data presented, the effects of these three factors on PSC sponge extraction efficiency were predicted using a second-order polynomial function: Y = 92.62 + 4.48A + 4.41B + 2.79C + 2.52AB + 0.91AC+ 1.06BC − 7.62A^2^ − 4.88B^2^ − 5.81C^2^. (Where Y represented PSC extraction efficiency, and A, B, and C were pepsin concentration, solid-liquid ratio, and hydrolysis time, respectively).

Table 2 succinctly presents the statistical significance of the quadratic model and its outcomes. Employing the model equation, three-dimensional response surfaces and contour plots were created, aiding in visualizing the relationship between PSC sponge extraction yield and extraction parameters (Appendix A). Notably, under specific conditions—pepsin concentration at 2067 U/g, a solid-liquid ratio of 1:83 g/mL, and a 10-h hydrolysis period—the maximum projected PSC sponge yield reached 93.76%, aligning closely with the expected value of 93.93%.

### 2.3. Collagen Characterization

In Figure 2A, the sodium dodecyl sulfate-polyacrylamide gel electrophoresis (SDS-PAGE) analysis of the extracted eel swim bladder collagen demonstrated that the characteristic bands of collagen remained intact and showed no apparent damage, indicating the preservation of collagen’s triple-helix conformation. Additionally, the α1 chain exhibited a larger relative molecular weight compared to the α2 chain, and their content ratio was approximately 2:1, further confirming that the PSC sponge conformed to the triple-helix structure characteristic of Type I collagen [(α1)2α2] [1,17]. In Figure 2B, the Ultraviolet (UV) absorption spectrum revealed an absorption peak at 235 nm for the PSC sponge sample, which is consistent with the UV absorption characteristics of type I collagen, further supporting the identification of the collagen type. Turning to Figure 2C, the Fourier transform infrared spectroscopy (FTIR) indicated the presence of amide A (3434 cm^−1^), amide B (2927 cm^−1^), amide I (1644 cm^−1^), amide II (1555 cm^−1^), and amide III (1241 cm^−1^), providing additional evidence of the collagen’s molecular composition [17]. Scanning electron microscopy (SEM) analysis, as shown in Figure 2D, revealed that the ultrastructure of collagen appeared as a multilayer aggregation, with collagen fibers interwoven into an irregular porous network structure, effectively preserving the integrity of the collagen structure [18]. Collectively, the above findings support the conclusion that the collagen extracted by the enzyme method is the type I collagen.

### 2.4. Animal Experiment

#### 2.4.1. Wound Healing Rate

The wound healing phenomenon in mice is depicted in Figure 3 In the control group, a blood scab initially formed on the wound surface, accompanied by noticeable redness and swelling. With the extension of time, the wound was gradually reduced in size, followed by the formation of a scab. Eventually, the scab naturally detached, and new hair started to grow in the surrounding area.

In contrast to the control group, the PSC sponge group exhibited milder redness and inflammation in the initial phase of wound healing. Furthermore, the PSC sponge group exhibited accelerated wound healing, complete regeneration of the skin, and vigorous hair growth. As depicted in Table 3, the PSC sponge group exhibited a notably higher skin wound healing rate compared to the control group, showing a statistically significant difference (*p* < 0.05). In conclusion, PSC sponges can provide effective protection for skin wounds and promote skin wound healing.

#### 2.4.2. Serum Antioxidant Activity

Compared with the control group, the results of serum biochemical indices in the PSC sponge group were shown in Figure 4. The expression activities of SOD, CAT, T-AOC, and GSH-Px in the PSC sponge-treated group were significantly increased (*p* < 0.05). Conversely, the levels of MDA, a marker of oxidative stress, were significantly decreased in the PSC sponge-treated group (*p* < 0.05). This suggests that PSC sponge has better antioxidant activity, possibly by preventing excessive oxidation to promote wound healing.

#### 2.4.3. Serum Inflammatory Factor Levels

As depicted in Figure 5, the PSC sponge group exhibited significantly reduced levels of IL-6, IL-1β, and TNF-α compared to the control group (*p* < 0.05). This observation indicates that PSC effectively inhibited the expression of inflammatory factors in the skin wounds of mice, thereby reducing the inflammatory response and promoting the healing process.

#### 2.4.4. Hematoxylin-Eosin (H&E) Staining

The histopathological results of the skin incision for both the control group and the PSC sponge group are presented in Figure 6. In the control group, the incision at 3 days (A_1_) showed a significant infiltration of inflammatory cells (a). However, on the 3rd day of the incision in the PSC sponge group (A_2_), a large amount of red membranous tissue covered the surface with only a few inflammatory cells present. By day 7, the control group showed no epidermal cell layer observed on the incision site (B_1_). In contrast, the PSC sponge group (B_2_) exhibited the formation of some epidermal cells (e) on the incision surface, accompanied by the development of granulation tissue in the dermis. Notably, the formation of skin appendages, such as sebaceous glands (b) and sweat ducts (c), was also observed in the PSC sponge group.

Two weeks later, the PSC sponge group (C_2_) displayed a significantly better formation of skin appendages (sebaceous gland and hair follicle, etc., b, d) in both the epidermis layer and dermis compared to the control group (C_1_). After 21 days, the control group (D_1_) showed 3 to 4 layers of epidermal cells (a). In contrast, the PSC sponge group (D_2_) exhibited the formation of stratified flat epithelium in the epidermis, with an obvious inner papillary layer (g) in the dermis. Additionally, there were abundant collagen fibers and blood vessels in the reticular layer, and the structure of sebaceous and sweat glands (f) closely resembled normal tissue. In conclusion, the HE staining of the incision skin indicated that the PSC sponge could effectively reduce bleeding and inflammation of the wound while promoting skin and appendage regeneration.

#### 2.4.5. Immuno-Histochemical Analysis of CD31

On the third day of the experiment, as shown in Figure 7A_1_,A_2_, there were no CD31 positive sites observed in the wound subcutaneous tissue of the control group and the PSC sponge group. On the 7th day of the experiment, as shown in Figure 7B_1_,B_2,_ and on the 14th day, as shown in Figure 7C_1_,C_2_, CD31 was positively expressed in both groups, with significantly higher expression in the PSC sponge group compared to the former. However, by the 21st day of the experiment, as indicated in Figure 7D_1_,D_2_, the expression of CD31 positive sites at this time was reduced compared to the 14th day, indicating a reduction in blood vessel formation in the wound healing process.

#### 2.4.6. Immuno-Histochemical Analysis of EGF

The results revealed distinct patterns of EGF expression during different stages of the experiment. On the third day of the study (Figure 8A_1_,A_2_), there was no significant positive expression of EGF in the wound tissues of both groups. However, by the 7th day (Figure 8B_1_,B_2_), a small amount of EGF positive expression was observed in the control group, while the PSC sponge group exhibited significantly higher expression, particularly in the surface cells. As the experiment progressed to the 14th day (Figure 8C_1_,C_2_), EGF expression reached its peak in both groups, In the control group, EGF expression was mainly localized to the newborn epidermis, while in the PSC sponge group, the expression of EGF in the epidermis and dermis, particularly in sebaceous glands and sweat glands, was more pronounced. On the 21st day (Figure 8D_1_,D_2_), EGF positive expression was notably reduced compared to the 14th day, and the wounds in the PSC sponge group had largely healed.

#### 2.4.7. Immuno-Histochemical Analysis of TGF-β1

On the 3rd day (Figure 9A_1_,A_2_), both the control group and the PSC sponge group showed positive expression of TGF-β1 in the wound tissue. However, the PSC sponge group exhibited more pronounced TGF-β1 expression compared to the control group. As the experiment progressed to the 7th day (Figure 9B_1_,B_2_) and the 14th day (Figure 9C_1_,C_2_), positive expression of TGF-β1 was observed in the epidermal cells and the dermis in both groups of tissues. Notably, the PSC sponge group demonstrated significantly higher TGF-β1 expression levels compared to the earlier time points, indicating an ongoing increase in TGF-β1 during the wound healing process. By the 21st day (Figure 9D_1_,D_2_), the expression of TGF-β1 positive sites in both groups decreased compared with that in 14 days suggesting a potential decline in TGF-β1 signaling role in the later stages of wound healing.

#### 2.4.8. Immuno-Histochemical Analysis of Type I Collagen

On day 3 (Figure 10A_1_,A_2_), positive expression sites of type I collagen were not prominently observed in either group. However, by days 7 (Figure 10B_1_,B_2_) and 14 (Figure 10C_1_,C_2_), there was a notable increase in the positive expression of type I collagen in the subcutaneous tissue. Importantly, the degree of positive expression was significantly higher in the PSC sponge group than in the control group, particularly in the dermis. As the experiment reached day 21 (Figure 10D_1_,D_2_), the expression of type I collagen was significantly reduced compared to that at day 14, and the reduction was more pronounced in the PSC sponge group.

## 3. Discussion

The global collagen market had a size of $1.15 billion in 2021 and is projected to reach $1.46 billion by 2026, with a compound annual growth rate (CAGR) of 4.90% during this period [19]. Therefore, our country has gradually paid attention to the application and research, and development of collagen. However, the process of aquatic animals yields 75% waste output, a substantial proportion of which, approximately 35%, is composed of fish swim bladder, skin, bones, and scales [20,21]. Notably, these byproducts are richly laden with collagen, indicating an untapped resource that could be judiciously harnessed. In this study, the response surface method was used to optimize the extraction process of PSC from an eel swim bladder, which greatly improved the extraction rate of collagen. The eel swim bladder PSC sponge was found to be type I collagen with a complete triple helix structure, and its morphology indicated a highly porous structure with very high porosity. 

Collagen, as an integral structural and functional constituent of the dermal extracellular matrix, exerts a pivotal influence in the process of wound healing [22]. Due to its unique perforated structure, high porosity, and unique biological property [23], Evaluations were made on serum inflammatory markers, and antioxidant levels, as well as through histological (H&E) and immune-histochemical analyses, juxtaposed against control groups. In instances of full-thickness wounds in ICR mice, the PSC porous sponge could effectively adsorb the wound exudate during the inflammatory stage while simultaneously facilitating an appropriate rate of water evaporation, thereby nurturing an optimal moist wound environment. Remarkably, PSC-treated groups exhibited significantly enhanced wound healing capability, increased fibroblast proliferation, and elevated collagen synthesis in comparison to the control groups.

Based on the preceding discoveries, the accelerated wound healing attributed to the PSC sponge may be ascribed to its distinctive structural attributes and notably porous matrices [24]. These attributes prompt the proliferation of fibroblasts and the synthesis of collagen, playing pivotal roles in the processes of re-epithelialization and vascularization during wound healing [25]. Furthermore, the biocompatibility of the PSC sponge is noteworthy, promoting the attachment and growth of fibroblasts, thereby facilitating the deposition and maturation of collagen [26]. Over time, collagen undergoes complete degradation and absorption within the wound, fostering the formation of fibroblasts and collagen fibers within the epidermal tissue. Consequently, the regeneration of the dermal layer in place of the wounds.

The results of serum detection in this study showed that the PSC sponge increased antioxidant levels, as evidenced by elevated activities of SOD, CAT, T-AOC, and GSH-Px. Conversely, it led to a reduction in MDA level, indicating enhanced antioxidant defense mechanisms and decreased oxidative stress. Additionally, the PSC sponge demonstrated the ability to inhibit the levels of inflammatory factors IL-6, IL-1β, and TNF-α, thereby reducing the inflammatory response in mouse skin wounds. As a result, wound healing was accelerated, and the regeneration of skin, hair, and accessory organs was promoted. Histopathological and immunohistochemical analyses demonstrate that the utilization of a PSC sponge exerts a favorable influence on neovascularization, prompting the proliferation of fibroblasts, synthesis of collagen, re-epithelialization, and the regeneration of skin structures. As a result, this notably augments the wound healing capacity when compared to control groups. Furthermore, owing to its three-dimensional characteristics and high porosity, the PSC sponge creates an appropriate milieu for the adhesion of fibroblasts within the skin.

Crucial components involved in the wound healing process include the vascular endothelial marker CD31, growth factors such as EGF and TGF-β, as well as type I collagen [27]. It has been validated that during the initial stages of wound healing, a substantial upsurge in the presence of CD31 and EGF-positive sites is discernible within the PSC sponge-treated group, accompanied by a conspicuous expression of type I collagen. This observation suggests that the PSC sponge holds the potential to stimulate the formation of new blood vessels, encourage fibroblast proliferation, promote enhanced collagen synthesis, and thus contribute to the advancement of tissue wound repair. In the latter stages, there was a gradual decline in the expression of CD31 and EGF, along with a substantial reduction in neovascularization. This is propitious for the reconfiguration of the epidermal structure and the growth of fibroblasts, thus reducing the formation of scar tissue. In addition, PSC sponges can promote the expression of TGF-β1 and accelerate the formation of granulation tissue, favoring wound healing. The subsequent reduction in its expression in the later stages can mitigate scar formation.

In this study, the PSC sponge was effectively crafted and assessed for its potential application as a wound dressing to facilitate skin regeneration. Application of PSC notably amplified wound healing capabilities, fibroblast proliferation, collagen synthesis, re-epithelialization, and the reconstitution of the dermal layer in vivo [28]. Research has demonstrated that the incorporation of plant extracts into collagen-based nano scaffolds was explored to enhance the efficacy of wound healing in diabetic conditions [29]. Besides, collagen exhibits versatility in blending with various polymers, such as bio-ceramics, carbon, and polymer materials, allowing for adjustments in properties like mechanical strength and antibacterial attributes [30]. Furthermore, Zheng et al. demonstrated that the combination of phenolic acid grafted chitosan and collagen in food packaging composite films enhances the antioxidant and antibacterial properties, while also improving mechanical strength, thermal stability, resistance to ultraviolet light, and water vapor barrier performance [31]. Composite hydrogels containing PSC sponge and chitooligosaccharides were found to show noteworthy anti-inflammatory activity and biocompatibility [32]. These findings underscore the promising prospects of PSC sponges in the field of biomedical materials.

## 4. Materials and Methods

### 4.1. Materials 

Fresh sea eels were purchased from Zhoushan Farmers’ Vegetable Market and pretreated before being stored at −20 °C. Sub-high molecular weight Marker (43–200 kDa) from Shanghai Institute of Life Sciences, Chinese Academy of Sciences; hydroxyproline (HYP), CAT, SOD, MDA, GSH-Px, T-AOX, IL-6, IL-1β, TNF-α determination kit from Nanjing Institute of Bioengineering; fish skin collagen standard from Beijing Solaibao Technology Co., Ltd. (Beijing, China); pepsin (1:15,000 U) from Beijing Asia-Pacific Hengxin Biotechnology Co., Ltd. (Beijing, China); anhydrous ethanol, n-butanol, Coomassie bright blue R-250, sodium hydroxide, glacial acetic acid, potassium bromide, 4% chloral hydrate, and sodium dodecyl sulfonate (SDS), 30% propylene amide all from Agent Company.

ICR male healthy mice, weighing 22–24 g, were procured from Zhejiang Hangzhou Ziyuan Experimental Animal Technology Co., Ltd. (Hangzhou, China). The mice were housed and raised in the experimental animal room of Zhejiang Ocean University (license number: SYXK (Zhejiang) 2019-0031).

### 4.2. Extraction of Fish Bladder Collagen

#### 4.2.1. Extraction Process of Fish Bladder Collagen

The extraction of fish bladder collagen was performed following the method described by Mo et al. [33]. The extraction process involved a series of sequential steps. Firstly, the fish bladder was thoroughly washed. Subsequently, the internal blood vessels and connective tissue were carefully removed. Next, a 0.1 mol/L NaOH solution was used to eliminate miscellaneous proteins. Then, a 10% n-butanol solution was employed for lipid removal. Following that, the fish bladder collagen was extracted by shaking the mixture at a constant temperature of 24 °C and 120 r/min using a 0.5 mol/L acetic acid solution containing pepsin. The extracted solution was then subjected to filtration, and the filtrate was collected. To further purify the collagen, dialysis with distilled water was carried out to achieve a pH of 7.0. Finally, the freeze-drying process was utilized to complete the extraction procedure.

#### 4.2.2. Determination of Collagen Extraction Rate

After processing the fish bladder according to the aforementioned method, the filtrate was collected as the sample. The hydroxyproline content in the sample was determined by measuring its absorbance using the HPY assay kit and referencing the standard curve. Following the guidelines in the literature [34], the hydroxyproline content and collagen extraction rate in the fish bladder were calculated using Equations (1) and (2), respectively.
(1)$$W_{1}\%=\frac{h\times 10^{6}\times 50}{m}\times 100$$
(2)$$W_{2}\%=\frac{m_{1}}{m_{2}\times W_{1}\times 11.1}\times 100$$

In the formula: *W*_1_ is the content of hydroxyproline/% in the sample; *h* is the mass concentration of hydroxyproline corresponding to the standard curve/(μg/mL); *m* is the mass/g of fish swim bladder when determining the content of hydroxyproline. *W*_2_ is the collagen extraction rate/%; *m*_1_ is the quality of the extracted collagen drying product/g; *m*_2_ is the quality/g of the fish swim bladder at the time of extraction; *W*_1_ is the hydroxyproline content/% in the sample; 11.1 is the content coefficient of hydroxyproline and collagen.

#### 4.2.3. Single-Factor Test

Based on the previous experiment [35], a single-factor test was performed to investigate the impact of enzyme addition on collagen extraction from the swim bladder of eel in group A, with enzyme concentrations of 1000, 1500, 2000, 2500, and 3000 U/g. Similarly, in group B, different solid-liquid ratios of 1:20, 1:40, 1:60, 1:80, and 1:100 g/mL were examined. Lastly, in group C, varying enzymatic digestion times of 4, 6, 8, 10, and 12 h were studied. The focus was specifically on assessing the effect of different conditions on the extraction rate of collagen.

#### 4.2.4. Response Surface Optimization Test

Utilizing the principles of the Box-Behnken design [36], three factors that displayed significant effects, namely enzyme addition (A), feed-to-liquid ratio (B), and enzymatic digestion time (C), were selected based on the results of the single-factor test. Employing a three-factor, three-level response surface analysis approach, we explored the interactions and optimized the conditions for collagen extraction. The specific test factors and corresponding levels are outlined in Appendix A.

### 4.3. Analysis of the Properties of PSC

#### 4.3.1. SDS-PAGE Gel Electrophoresis

Following the provided guidelines, 8% acrylamide isolation gels and 5% acrylamide concentration gels were meticulously prepared. Electrophoresis was carried out using Tris-glycine buffer, and the freeze-dried PSC sponge sample was dissolved. After adding the loading buffer, boil for 5 min, then load 10 μL of the sample onto the gel. ADC constant current power supply with a voltage of 180 V and a current of 110 mA was employed for electrophoresis, lasting approximately 1–2 h. Subsequently, the gel was carefully retrieved and subjected to Coomassie bright blue dye staining for 30 min [20]. The decolorization process was conducted until the blue background lightened or disappeared, and clear bands emerged. After thorough deionized water rinsing, the experimental outcomes were meticulously recorded.

#### 4.3.2. UV Absorption Spectrum

The freeze-dried PSC sponge was dissolved using 0.1 mol/L acetic acid as the solvent, generating a 0.2 mg/mL PSC sponge solution for testing purposes [37]. During the blank control utilizing the 0.1 mol/L acetic acid solution, the ultraviolet spectrometer scan detected the characteristic absorption peak of the solution within the wavelength range of 190~400 nm.

#### 4.3.3. FTIR Spectroscopy 

Two milligrams of collagen extracted from the sea eel swim bladder (diluted at a ratio of 1:100) was mixed with dried KBr powder in a dry and clean agate mortar, ground unidirectionally [38], and subsequently pressed into tablets using a mold. The FTIR analysis involved recording the spectrum over a wave number range of 400~4000 cm^−1^ at a resolution of 4 cm^−1^ with 40 scans accumulated.

#### 4.3.4. SEM Ultrastructure

Following standard scanning electron microscope procedures [39], the collagen samples were weighed, compacted, freeze-dried, fixed, and thinly sliced. The dried specimens were gold-sputtered under vacuum conditions to create samples for scanning electron microscopy observation at ×100, ×250, ×1000, and ×2500 magnifications.

### 4.4. Animal Experiments

#### 4.4.1. Skin Wound Healing in ICR Mice

The experiment on wound healing was executed in the subsequent manner: Male ICR mice, weighing 22–24 g, were domiciled in a pathogen-free habitat and furnished with a standardized diet. Before initiation, ICR mice (*n* = 26) underwent anesthesia through intraperitoneal administration of 4% chloral hydrate (0.1 mL/10 g). Square skin wounds measuring 1 cm in side length were introduced on the dorsum of the mice, positioned 1.5 cm away from the spine [40,41]. Segregated into two groups with equal allocation, the control group was administered a 0.9% saline solution [12,42]. In contrast, the wounds of the treated group were subsequently covered with PSC sponge dressings to prevent potential infections. Dressings were replaced every two-day interval, and the mice were nurtured in cages. On postoperative days 3, 7, 14, and 21, comprehensive assessments of wound morphology were performed, and images of the skin wounds were documented. After the collection of whole blood from the ocular region, euthanasia was carried out. At specified intervals encompassing the 3rd, 7th, 14th, and 21st-day post-wound creation, wound tissues were excised from three mice in each group, and granulation tissues were extracted. Utilization of the Image J software facilitated the quantification of wound area for each grouping, thereby enabling computation of the wound healing rate for distinct time points as per the ensuing formula: Wound healing rate = (Initial wound area − Unhealed wound area)/Initial wound area × 100%.

#### 4.4.2. Determination of Serum Antioxidant Activity Index 

After collecting the mice’s blood, it was allowed to stand in an EP tube at room temperature (25 °C) for 1 to 2 h before being centrifuged at 4 °C and 4000 r/min for 15 min. Subsequently, the serum was carefully extracted, and the activities of SOD, CAT, T-AOC, GSH-Px, and MDA levels were assessed following the instructions provided with the respective assay kits.

#### 4.4.3. Detection of Serum Inflammatory Factor Levels

Mice serum levels of the inflammatory factors IL-6, IL-1β, and TNF-α were determined using an enzyme-linked immunosorbent assay (ELISA) kit, following the manufacturer’s instructions.

#### 4.4.4. Histopathological Examination

Samples collected on postoperative days 3, 7, 14, and 21 were fixed using a 4% paraformaldehyde solution for 24 h. Subsequently, the tissues underwent paraffin embedding and were sectioned into slices of 5 µm thickness, which were subsequently subjected to histopathological examination employing the H&E staining technique. The sections were then observed utilizing a conventional light microscope. The histopathological assessment focused on evaluating re-epithelialization, collagen deposition, and the presence of inflammatory cell infiltration [43].

#### 4.4.5. Immuno-Histochemical Examinations

The above-mentioned embedded wax block slices were carried out through the immuno-histochemical (SP) method to carry out antigen-antibody reaction, and the positive expression of CD31, EGF, TGF-β1, and type I collagen respectively. Microscopic observation and photography were conducted to record the results [44]. The number of new blood vessels is expressed by counting the number of CD31 tubes. Image J software statistically analyzes the positive expression of TGF-β, EGF, and type I collagen in a certain area [45]. The expression rate was calculated as the brown particle expression area divided by the total area.

### 4.5. Data Statistical Analysis

All the results were reproduced at least three independent experiments and reported as mean values ± standard derivation (SD). Statistical significance (*p* < 0.05) between diverse groups was assessed through Student’s *t*-test and one-way analysis of variance (ANOVA), accompanied by Tukey’s post hoc multiple comparison test. All statistical computations were executed employing SPSS 23.0 software.

## 5. Conclusions

In summary, this study aimed to extract and characterize collagen from sea eel swim bladders and develop a collagen sponge for skin wound healing. Utilizing Response Surface Methodology (RSM), we achieved a remarkable collagen extraction yield of 93.76% under specific conditions: pepsin concentration of 2000 U/g, hydrolysis time of 10 h, and a solid-liquid ratio of 1:80 g/mL. The extracted collagen was identified as type I collagen with a distinctive fibrous structure. When applied as a wound dressing in mice, the collagen sponge exhibited exceptional wound healing properties. It effectively reduced inflammation, accelerated tissue repair, and promoted skin regeneration. The sponge also upregulated key factors like CD31, EGF, TGF-β1, and type I collagen, facilitating tissue regeneration while minimizing scar formation. Moreover, it demonstrated antioxidant properties, enhancing antioxidant levels and reducing oxidative stress and inflammation markers. Histological and immunohistochemical analyses further confirmed its role in promoting tissue repair processes. These findings underscore the potential of the PSC sponge as a valuable biomedical material for wound healing applications. Future research may explore its compatibility with additives and polymers to enhance its properties for a wide range of medical and pharmaceutical applications. Furthermore, in upcoming studies, we plan to incorporate a control group consisting of inert materials to expand our research scope.

## Figures and Tables

**Figure 1 marinedrugs-21-00525-f001:**
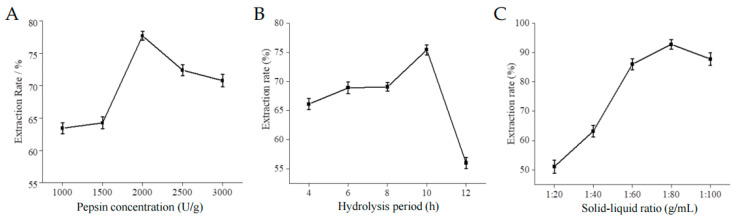
Effect of pepsin concentration (**A**), hydrolysis time (**B**), and solid-liquid ratio (**C**) on collagen extraction rate.

**Figure 2 marinedrugs-21-00525-f002:**
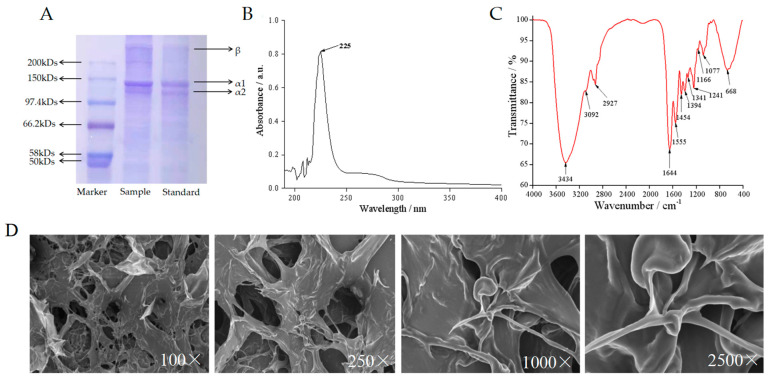
SDS-PAGE electrophoresis pattern (**A**); UV absorption spectrum (**B**); FTIR spectroscopy (**C**); SEM ultrastructure at 100×, 250×, 1000×, 2500× (**D**) of PSC from swim bladders of sea eel.

**Figure 3 marinedrugs-21-00525-f003:**
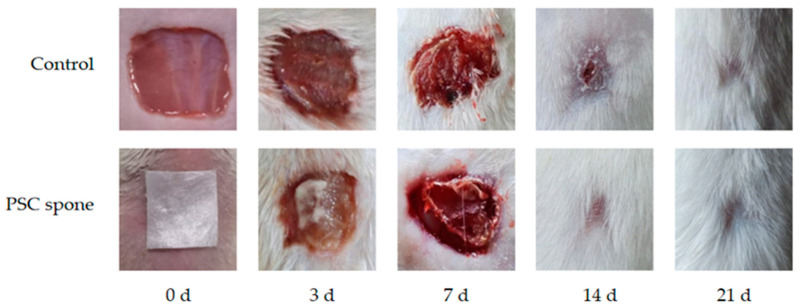
Captured representative images of wound healing with PSC sponge on days 0, 3, 7, 14, and 21.

**Figure 4 marinedrugs-21-00525-f004:**
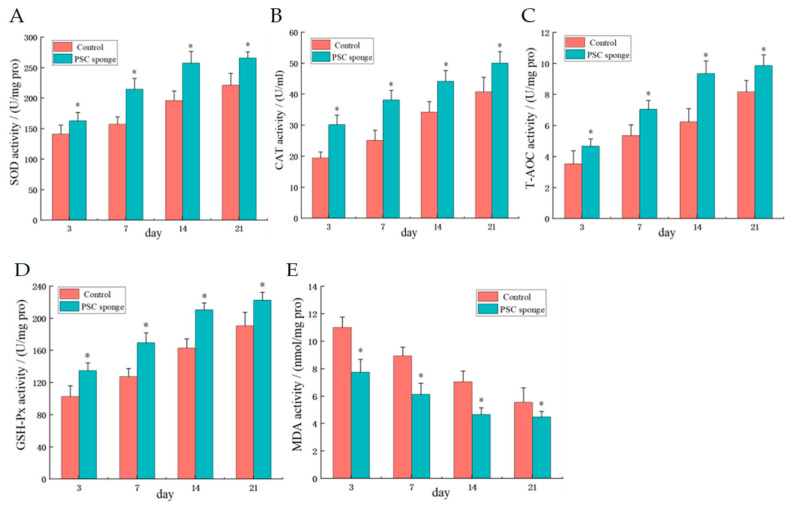
Effects of PSC sponge on serum antioxidant activity of SOD (**A**), CAT (**B**), T-AOC (**C**), GSH-Px (**D**), and MDA (**E**). * indicates significant difference compared with control group (*p* < 0.05).

**Figure 5 marinedrugs-21-00525-f005:**
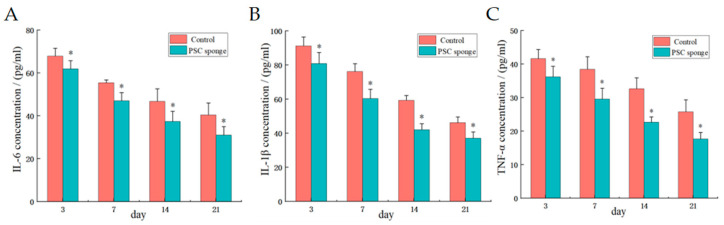
Effects of PSC sponge on serum inflammatory factor levels of IL-6 (**A**), IL-1β (**B**), and TNF-α (**C**). * indicates significant difference compared with control group (*p* < 0.05).

**Figure 6 marinedrugs-21-00525-f006:**
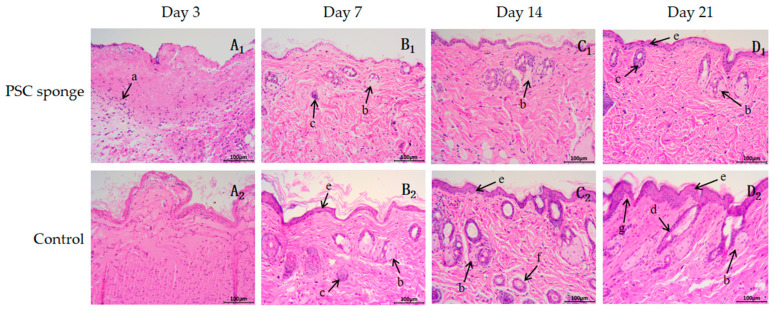
Histological analysis of H&E staining wounded tissues (Magnification, ×200). (**A_1_**–**D_1_**) are the control group on days 3, 7, 14, and 21; (**A_2_**–**D_2_**) are PSC sponge dressing group on days 3, 7, 14, and 21. a, inflammatory cell; b, sebaceous gland; c, sweat duct; d, hair follicle; e is the epidermal cell; f, secretory part of the sweat gland; g, papillary layer.

**Figure 7 marinedrugs-21-00525-f007:**
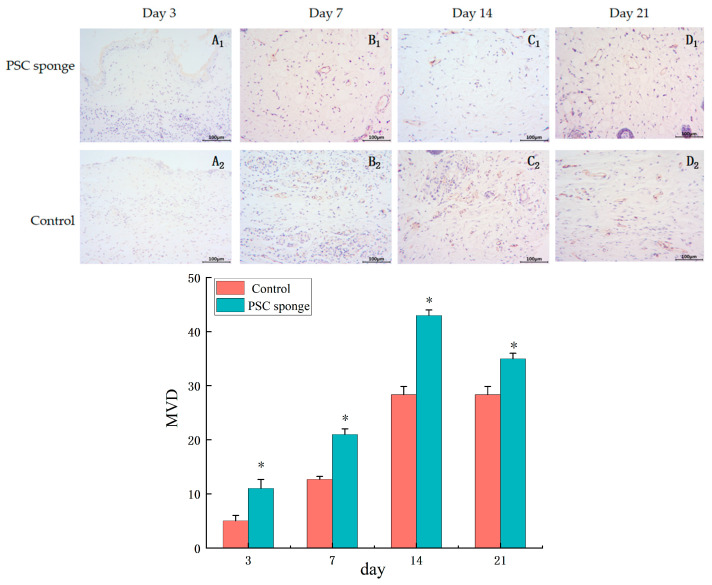
Immuno-histochemical analysis of CD31 expression in wounded tissues (Magnification, ×200). (**A_1_**–**D_1_**) are the control group on days 3, 7, 14, and 21; (**A_2_**–**D_2_**) are the PSC sponge dressing group on days 3, 7, 14, and 21. The histogram summarizes the microvessel density (MVD), assessed through immunohistochemical staining for CD31. * indicates significant difference compared with control group (*p* < 0.05).

**Figure 8 marinedrugs-21-00525-f008:**
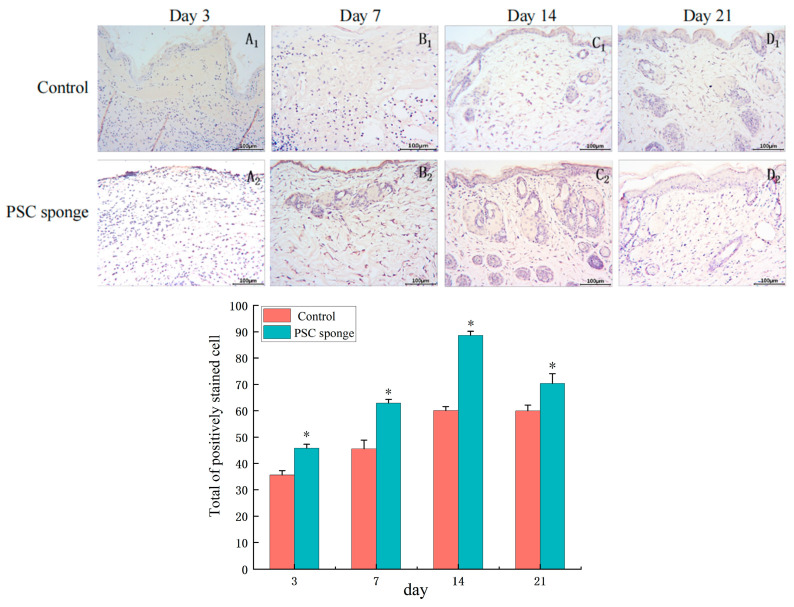
Immuno-histochemical analysis of EGF expression in wounded tissues (Magnification, ×200). (**A_1_**–**D_1_**) are the control group on days 3, 7, 14, and 21; (**A_2_**–**D_2_**) are PSC sponge dressing group on days 3, 7, 14, 21. The histogram displays the collective count of positively stained cells of EGF in the dermal tissue each respective group. * indicates significant difference compared with control group (*p* < 0.05).

**Figure 9 marinedrugs-21-00525-f009:**
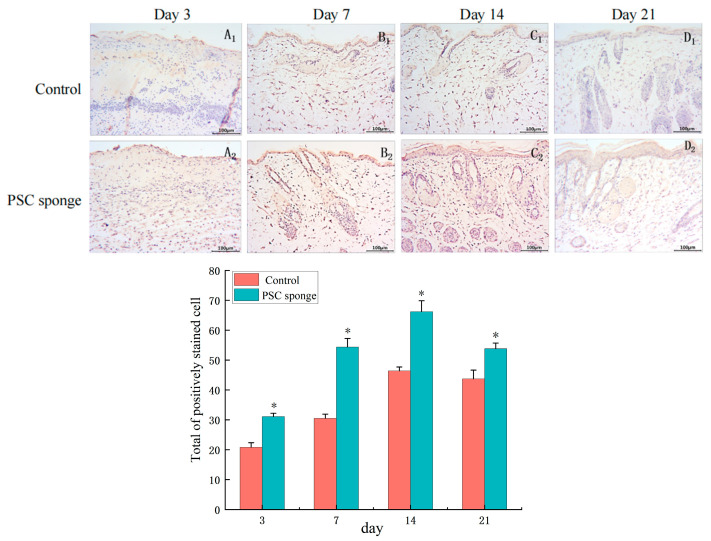
Immuno-histochemical analysis of TGF-β1 expression in wounded tissues (Magnification, ×200). (**A_1_**–**D_1_**) are the control group on days 3, 7, 14, and 21; (**A_2_**–**D_2_**) are the PSC sponge dressing group on days 3, 7, 14, and 21. The histogram displays the collective count of positively stained cells of TGF-β1 in the dermal tissue each respective group. * indicates significant difference compared with control group (*p* < 0.05).

**Figure 10 marinedrugs-21-00525-f010:**
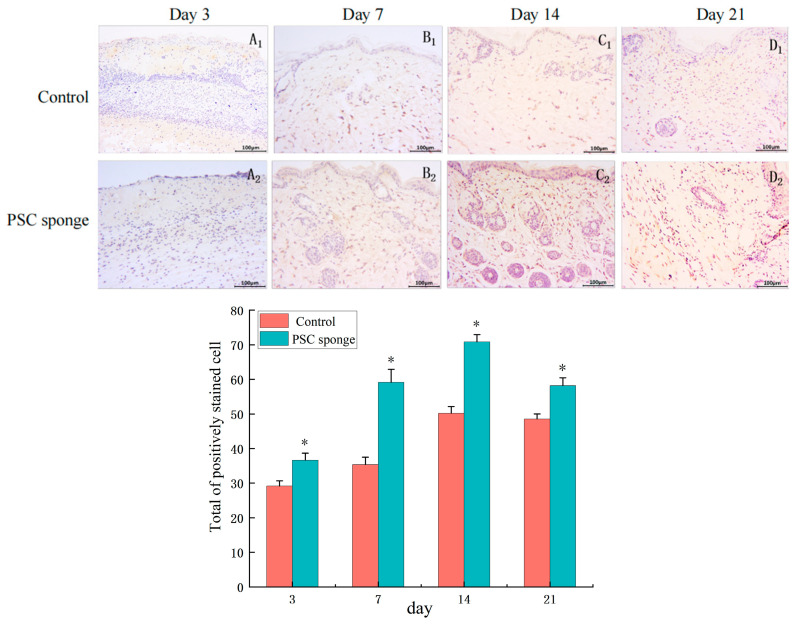
Immuno-histochemical analysis of type I collagen expression in wounded tissues (Magnification, ×200). (**A_1_**–**D_1_**) are the control group on days 3, 7, 14, and 21; (**A_2_**–**D_2_**) are the PSC sponge dressing group on days 3, 7, 14, 21. The histogram displays the collective count of positively stained cells of type I collagen in the dermal tissue each respective group. * indicates significant difference compared with control group (*p* < 0.05).

**Table 1 marinedrugs-21-00525-t001:** Box-Behnken experimental design and results.

Serial Number	Enzyme Concentration A/(U/g)	Solid-Liquid Ratio B/(g/mL)	Hydrolysis Time C/h	Extraction Rate Y/%
1	0	0	0	92.06
2	0	1	−1	82.63
3	−1	1	0	76.06
4	0	1	1	91.70
5	1	1	0	91.32
6	0	0	0	92.88
7	−1	0	−1	74.15
8	0	−1	−1	74.26
9	0	−1	1	79.09
10	1	0	−1	80.03
11	1	0	1	86.03
12	1	−1	0	79.12
13	0	0	0	92.91
14	−1	0	1	76.53
15	−1	−1	0	73.95

**Table 2 marinedrugs-21-00525-t002:** Analysis of variance of the regression model.

Source	Sum of Squares	df	Mean Square	F-Value	*p*-Value
Model	782.84	9	86.98	33.25	0.0006
A	162.72	1	162.72	62.20	0.0005
B	155.58	1	155.58	59.47	0.0006
C	60.50	1	60.50	23.13	0.0048
AB	25.40	1	25.40	9.71	0.0264
AC	2.79	1	2.79	1.07	0.3492
BC	4.49	1	4.49	1.72	0.2469
A^2^	212.47	1	212.47	81.22	0.0003
B^2^	89.41	1	89.41	34.18	0.0021
C^2^	123.18	1	123.18	47.08	0.0010
Residual	13.08	5	2.62		
Lack of Fit	12.62	3	4.21	18.08	0.0529
Pure Error	0.47	2	0.23		
Cor Total	795.92	14			
R^2^					0.9836
Adj R^2^					0.9540

Note: *p* < 0.01 indicates that the effect is extremely significant, 0.01 < *p* < 0.05 is significant, and *p* > 0.05 indicates that the effect is not significant.

**Table 3 marinedrugs-21-00525-t003:** Percentage of wound contraction recorded of the control and PSC sponge groups at different time points.

Groups	Wound Healing (%)			
	3 Days	7 Days	14 Days	21 Days
Control	37.54 ± 1.07	52.42 ± 0.76	93.05 ± 0.49	97.81 ± 0.20
PSC sponge	45.85 ± 0.74 *	63.19 ± 0.68 *	95.41 ± 0.33 *	99.14 ± 0.17 *

Note: Data are expressed as the mean ± SD, * indicates significant difference compared with control group (*p* < 0.05).

## Data Availability

Relevant information has been added to the article.

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
