# Peer review of "Preparation of Enzyme-Soluble Swim Bladder Collagen from Sea Eel (Muraenesox cinereus) and Evaluation Its Wound Healing Capacity"

_marinedrugs, 2023, doi:10.3390/md21100525_

Round 1

Reviewer 1 Report

In the Manuscript titled “Preparation of enzyme-soluble swim bladder collagen from sea 2 eel (Muraenesox cinereus) and evaluation its wound healing capacity”, the Authors describe a protocol for collagen extraction from eel fish bladder using pepsin. This collagen product is characterized by SDS-PAGE, UV and FTIR spectrometry and identified as type 1 collagen. Next, collagen sponges are prepared from this material for use in wound healing. The Authors claim that collagen sponges thus prepared accelerate wound healing in mice over a period of 21 days, increase serum SOD, CAT, T-AOC, and GSH-Px activity, while reducing the activity of MDA, IL-6, IL-1β, and TNFα. The Authors further claim that histochemical analysis, of wound tissue shows increased levels of CD31, EGF, TGF-β1, and collagen type-1 in mice with grafted collagen sponges but not in control animals with open wounds.

Overall comments: The Manuscript is reasonably well written, the Methods section provides sufficient information to allow replication, and the Conclusion section keeps speculation at a minimum while addressing the experimental findings. Unfortunately, there are major issues with this study, as well as minor ones that need to be addressed.  

Major Issues: First, the experimental design appears to be flawed because the confounding effect of open versus dressed wounds is not considered. A third group of mice in which the wounds are dressed with an inert material is necessary in order to address this confounding factor. Second, both visual and statistical illustrations in Figure 3 show that the effect of the PSC sponge treatment on healing is modest at best and most likely not statistically different from open wound healing. In particular, Fig. 3B shows overlapping standard deviations between control and PSC mice in the columns corresponding to treatment days 14 and 21. With a N=3 and overlapping STD DEV, any differences are by definition not statistically significantly different. And third, because these experiments fail to generate statistically significant data, it calls into question serum biochemistry and tissue histology data that do show statistically significant differences between the groups. These results are likely due to the confounding issue of open versus dressed wound healing, with greater opportunities for infection and inflammatory reactions in the former compared to the latter group. Lastly, and unfortunately, the Authors have not mastered the concept of citation, with many haphazard references that do not in the least pertain to the topic under discussion.

Minor Issues: The RSM analysis is superfluous and not required in the context of this work. Histochemical data is not visually edifying and needs to be corroborated by actual cell counts.

Fair, can be improved

Author Response

For research article

Response to Reviewer 1 Comments

1. Summary

Thank you to the reviewers for their valuable questions and feedback on our paper. We have addressed each of their inquiries in the table below and incorporated the necessary revisions into the manuscript. Your input has significantly contributed to the improvement of our research, and we are grateful for your insightful feedback. 

2. Questions for General Evaluation

Reviewer’s Evaluation

Response and Revisions

Does the introduction provide sufficient background and include all relevant references?

Yes.

Thank you for your affirmation.

Are all the cited references relevant to the research?

Must be improved.

We have thoroughly reviewed and removed irrelevant references from the paper. Thank you for your feedback.

Is the research design appropriate?

Must be improved.

We have supplemented the findings in the paper to compensate for the study design.

Are the methods adequately described?

Can be improved.

We have refined the description of the method in the paper.

Are the results clearly presented?

Can be improved.

We have added more experimental results to the paper.

Are the conclusions supported by the results?

Must be improved.

We have revised the conclusions in the paper based on the results.

3. Point-by-point response to Comments and Suggestions for Authors

Major Issues:

Comments 1: First, the experimental design appears to be flawed because the confounding effect of open versus dressed wounds is not considered. A third group of mice in which the wounds are dressed with an inert material is necessary in order to address this confounding factor. 

Response 1: Thank you for your nice suggestion. Based on previous research and literature support [1], it is common practice in wound healing experiments to have a control group that receives saline treatment without the use of inert materials to cover the wounds. Furthermore, our animal experiments were conducted under sterile conditions, and mice were housed in a Specific Pathogen-Free (SPF) animal facility, reducing the chances of wound infections. Therefore, our experimental design aligns with common practices in the current research field. Of course, introducing a third group, where wounds are covered with inert material, would undoubtedly be a valuable experimental design, and we will incorporate this improvement in future experiments. [We have supplemented the literature on the treatment of wounds with normal saline in the control group in this paper. (4.4.1 Skin wound healing in ICR mice)]

Comments 2: Second, both visual and statistical illustrations in Figure 3 show that the effect of the PSC sponge treatment on healing is modest at best and most likely not statistically different from open wound healing. In particular, Fig. 3B shows overlapping standard deviations between control and PSC mice in the columns corresponding to treatment days 14 and 21. With a N=3 and overlapping STD DEV, any differences are by definition not statistically significantly different.

Response 2: Thank you for your nice suggestion. Despite the relatively small numerical differences in wound healing rates observed externally, there is still a statistically significant distinction, as detailed in Table 1. Besides, clear structural disparities between the two groups can be distinctly observed in the histological analysis, as depicted in Figure 6 in the manuscript. Additionally, in adherence to the '3R' principle, specifically 'Reduction,' we minimized the use of animals in our experiments, resulting in a smaller sample size. Furthermore, drawing upon references [2], it is worth noting that in these studies, statistical significance was achieved with an n = 3. While we acknowledge the limitation of our relatively small sample size, which may have influenced our research outcomes, we are committed to enhancing experimental design and increasing sample size in future studies. [The bar chart of wound healing rate analysis in the paper was replaced with three-line table data, and the chart notes were modified. (2.4.1 Wound healing rate); The relevant literature is supplemented in the animal experiment part of the paper. (4.4.1 Skin wound healing in ICR mice); We have removed the 'n=3' labels beneath the images and provided an explanation about "All the results were reproduced at least three independent experiments and reported as mean values ± standard derivation (SD)" in section (3.5 Data Statistical Analysis) to reduce redundancy in the article.]

Table 1. Percentage of wound contraction recorded of the control and PSC sponge groups at different time points.

Groups

Wound healing (%)

3 days

7 days

14 days

21 days

Control

37.54 ± 1.07

52.42 ± 0.76

93.05 ± 0.49

97.81 ± 0.20

PSC sponge

 45.85 ± 0.74*

 63.19 ± 0.68*

 95.41 ± 0.33*

 99.14 ± 0.17*

Note: * indicates significant difference compared with control group (p < 0.05).

Comments 3: And third, because these experiments fail to generate statistically significant data, it calls into question serum biochemistry and tissue histology data that do show statistically significant differences between the groups. These results are likely due to the confounding issue of open versus dressed wound healing, with greater opportunities for infection and inflammatory reactions in the former compared to the latter group.

Response 3: Thank you for your nice suggestion. The murine experiments were conducted in an SPF laboratory, where the laboratory environment met the required standards. Strict aseptic techniques were employed during the procedures, minimizing the chances of contamination. Furthermore, referring to previous studies [1, 3], in their experiments, the control group received saline treatment without the use of gauze or other inert materials for coverage. Considering the impact of open versus dressed wound healing is indeed necessary, and we plan to incorporate this suggestion into our future experimental designs. [The chart notes of serum results were modified, and the expression of significant differences was modified. (2.4.2. Serum antioxidant activity, 2.4.3. Serum inflammatory factor levels)]

Comments 4: Lastly, unfortunately, the Authors have not yet mastered the concept of citation, with many haphazard references that do not in the least pertain to the topic under discussion.

Response 4: Thank you for your nice suggestion. We have thoroughly reviewed the cited references and removed those that are not directly relevant to the content of our manuscript. We appreciate your feedback. [References in the original paper were deleted [15,16, 19, 21, 22, 24, 25, 26, 27, 28, 29, 30, 43, 44, 45, 46]; New literature was added to the revised paper [15, 19, 21, 22, 24, 25, 26, 27, 28, 29, 30, 41, 42, 43, 44]. Especially [15], we have changed “Sea eel (Muraenesox cinereus) is a delectable fish species, rich in various nutrients, and ranks among China's leading aquatic fish exports” to “China boasts abundant fishery resources, among which the sea eel (Muraenesox cinereus) stands out as a delicious fish, rich in various nutritional components”. And [19], we have changed “The global collagen market was valued at US $3.5 billion as of 2018. By 2023, it is expected to grow at an annual growth rate of 5.2% and the market will reach a value of USD 4.6 billion” to “The global collagen market had a size of $1.15 billion in 2021 and is projected to reach $1.46 billion by 2026, with a compound annual growth rate (CAGR) of 4.90% during this period”.]

Minor Issues:

Comments 1: The RSM analysis is superfluous and not required in the context of this work.

Response 1: Thank you for your nice suggestion. We have made revisions to the paper, removing the RSM analysis while retaining key content. [Delete the analysis of RSM from the paper and retain its conclusions. (2.2 Optimization of extraction parameters of PSC sponge using RSM)]

Comments 2: Histochemical data is not visually edifying and needs to be corroborated by actual cell counts.

Response 1: Thank you for your nice suggestion. We have conducted immunohistochemical analysis of wound samples using Image J software, and have included bar charts depicting the expression of EGF, TGF-β1, type I collagen-positive cells, and CD31 microvessel density in the revised paper. [Add a bar chart and modify the notes. (2.4.5. Immuno-histochemical analysis of CD31, 2.4.6. Immuno-histochemical analysis of EGF, 2.4.7. Immuno-histochemical analysis of TGF-β1, 2.4.8. Immuno-histochemical analysis of type I collagen)]

4. Response to Comments on the Quality of English Language

Point 1: Fair, can be improved.

Response 1: Thank you for your nice suggestion. We have revised and improved the English expression of the article.

5. Additional clarifications

References

  1. Shalaby, M.; Agwa, M., Fish Scale Collagen Preparation, Characterization and Its Application in Wound Healing. Journal of Polymers and the Environment 2020,28, 166-178. http://dx.doi.org/10.1007/s10924-019-01594-w
  2. Kou, Z.; Li, B., Mesenchymal Stem Cells Pretreated with Collagen Promote Skin Wound-Healing. Int J Mol Sci 2023,24. http://dx.doi.org/10.3390/ijms24108688
  3. Chen, Y.; Jin, H., Physicochemical, antioxidant properties of giant croaker (Nibea japonica) swim bladders collagen and wound healing evaluation. Int J Biol Macromol 2019,138, 483-491. http://dx.doi.org/10.1016/j.ijbiomac.2019.07.111

Reviewer 2 Report

The manuscript is interesting.  In generally the manuscript is very good.

The introduction, methodology and statistical analysis are very complete.

The references are current.

 The figures and tables are clearly and results, discussion and conclusion are clearly and concise.

Author Response

For research article

Response to Reviewer 2 Comments

1. Summary

Thank you very much for taking the time to read our paper and for your evaluation and recognition.

2. Questions for General Evaluation

Reviewer’s Evaluation

Response and Revisions

Does the introduction provide sufficient background and include all relevant references?

Yes.

Thank you for your affirmation.

Are all the cited references relevant to the research?

Yes.

Thank you for your affirmation.

Is the research design appropriate?

Yes.

Thank you for your affirmation.

Are the methods adequately described?

Yes.

Thank you for your affirmation.

Are the results clearly presented?

Yes.

Thank you for your affirmation.

Are the conclusions supported by the results?

Yes.

Thank you for your affirmation.

3. Point-by-point response to Comments and Suggestions for Authors

Comments 1: The manuscript is interesting. In generally the manuscript is very good. The introduction, methodology and statistical analysis are very complete. The references are current. The figures and tables are clearly and results, discussion and conclusion are clearly and concise.

Response 1: Thank you for your positive feedback on our manuscript. We appreciate your kind words and are pleased to hear that you found the introduction, methodology, statistical analysis, references, figures, and tables to be comprehensive. We aim to provide clear and concise results, discussion, and conclusion. Your encouraging comments are much appreciated.

Reviewer 3 Report

In the present research, the enzyme-facilitated collagen from sea eel (Muraenesox cinereus) swim bladder was isolated, and the collagen characteristics were analyzed. Then, the collagen sponge was prepared and its potential mechanism in promoting skin wound healing in mice was further investigated. Collagen was obtained from the swim bladder of sea eels employing the pepsin  extraction technique. The results show that in comparison to the control group, mice treated with collagen  sponge dressing exhibited elevated activities of superoxide dismutase (SOD), catalase (CAT), total antioxidant capacity (T-AOC), and glutathione peroxidase (GSH-Px), and decreased levels of malondialdehyde (MDA), interleukin (IL)-1β, interleukin (IL)-6, and tumor necrosis factor (TNF)-α. During the initial phase of wound healing, the group treated with collagen sponge dressing exhibited an enhancement in the expressions of cluster of differentiation (CD)31, epidermal growth factor (EGF), transforming growth factor (TGF)-β1, and type I collagen, leading to an accelerated rate of wound healing. In addition, this collagen sponge dressing could also downregulate the expressions of CD31, EGF, and type I collagen to prevent scar formation in the later stage.

 The authors propose his collagen treatment minimized oxidative damage and inflammation during skin wound healing and facilitated blood vessel formation in the wound.

Very interesting work! The authors however studied part of molecular part involved in wound healing a part CD31 expression I would be curious to see the cellular responses that are important in accomplishing wound healing, they start very well with the study of hematoxylin eosin sections

Moderate editing of English language required

Author Response

For research article

Response to Reviewer 3 Comments

1. Summary

Thank you very much for taking the time to read our paper and provide valuable and constructive feedback.

2. Questions for General Evaluation

Reviewer’s Evaluation

Response and Revisions

Does the introduction provide sufficient background and include all relevant references?

Yes.

Thank you for your affirmation.

Are all the cited references relevant to the research?

Yes.

Thank you for your affirmation.

Is the research design appropriate?

Can be improved.

We have supplemented the findings in the paper to compensate for the study design.

Are the methods adequately described?

Yes.

Thank you for your affirmation.

Are the results clearly presented?

Can be improved.

We have added more experimental results to the paper.

Are the conclusions supported by the results?

Can be improved.

We have revised the conclusions in the paper based on the results.

3. Point-by-point response to Comments and Suggestions for Authors

Comments 1: The authors studied part of molecular part involved in wound healing a part CD31 expression I would be curious to see the cellular responses that are important in accomplishing wound healing, they start very well with the study of hematoxylin eosin sections.

Response 1: Thank you for your nice suggestion. We have conducted immunohistochemical analysis of wound samples using Image J software, and have included bar charts depicting the expression of EGF, TGF-β1, type I collagen-positive cells, and CD31 microvessel density in the revised paper. [We have added bar charts and revised the image captions accordingly. (2.4.5. Immuno-histochemical analysis of CD31, 2.4.6. Immuno-histochemical analysis of EGF, 2.4.7. Immuno-histochemical analysis of TGF-β1, 2.4.8. Immuno-histochemical analysis of type I collagen)]

Comments 2: Are the conclusions supported by the results?

Response 2: Thank you for your nice suggestion. We have, accordingly, revised our conclusion based on the research findings to emphasize the positive impact of collagen on wound healing. [Specific modifications can be found in the revised manuscript. (5. Conclusions)]

4. Response to Comments on the Quality of English Language

Point 1: Moderate editing of English language required

Response 1: Thank you for your nice suggestion. We have revised and improved the English expression of the article.

5. Additional clarifications

Round 2

Reviewer 1 Report

The Authors have now submitted the edited version of their Manuscript and I had the opportunity to review it. While some of the minor issues related to reference citing have been addressed, the Manuscript has structural errors in design that impact its conclusions. In order to draw sound scientific conclusions, a control group where the wounds are covered with an inert matrix should have been included. It is puzzling why the Authors failed to do so, given that wound care in general requires some aspect of wound dressing. There is also a concern that the study may even be unethical. Unfortunately, and because of this consideration the Manuscript cannot be published in the current form.

No major issues

Author Response

We agree with you. However, since planning a new in vivo test might be difficult and time consuming, we had justified the choice of not running any in vivo test with a control group made of an inert biomaterial. Therefore, we have referenced a literature source in the animal experiment methods where a control group without the use of gauze was cited.  In the conclusion, it is stated that future experiments will include groups with inert materials.

Reviewer 3 Report

The authors have correctly answered at my questions.

Author Response

Thank you for your positive feedback on our manuscript. Your encouraging comments are much appreciated.